# Predictors of Recurrent Laboratory-Confirmed Symptomatic SARS-CoV-2 Infections in a Cohort of Healthcare Workers

**DOI:** 10.3390/vaccines11030626

**Published:** 2023-03-10

**Authors:** Xóchitl Trujillo, Oliver Mendoza-Cano, Mónica Ríos-Silva, Miguel Huerta, José Guzmán-Esquivel, Verónica Benites-Godínez, Agustin Lugo-Radillo, Jaime Alberto Bricio-Barrios, Martha I. Cárdenas-Rojas, Eder Fernando Ríos-Bracamontes, Vannya Marisol Ortega-Macías, Valeria Ruiz-Montes de Oca, Efrén Murillo-Zamora

**Affiliations:** 1Centro Universitario de Investigaciones Biomédicas, Universidad de Colima, Av. 25 de Julio 965, Col. Villas San Sebastián, Colima 28045, Mexico; 2Facultad de Ingeniería Civil, Universidad de Colima, km. 9 Carretera Colima-Coquimatlán, Coquimatlán 28400, Mexico; 3Centro Universitario de Investigaciones Biomédicas, CONACyT—Universidad de Colima, Av. 25 de Julio 965, Col. Villas San Sebastián, Colima 28045, Mexico; 4Unidad de Investigación en Epidemiología Clínica, Instituto Mexicano del Seguro Social, Av. Lapislázuli 250, Col. El Haya, Villa de Álvarez 28984, Mexico; 5Coordinación de Educación en Salud, Instituto Mexicano del Seguro Social, Calzada del Ejercito Nacional 14, Col. Fray Junípero Serra, Tepic 63160, Mexico; 6Unidad Académica de Medicina, Universidad Autónoma de Nayarit, Ciudad de la Cultura Amado Nervo, Tepic 63155, Mexico; 7CONACyT—Facultad de Medicina y Cirugía, Universidad Autónoma Benito Juárez de Oaxaca, Ex Hacienda Aguilera S/N, Carr. a San Felipe del Agua, Oaxaca 68020, Mexico; 8Facultad de Medicina, Universidad de Colima, Av. Universidad 333, Col. Las Víboras, Colima 28040, Mexico; 9Departamento de Medicina Interna, Hospital General de Zona No. 1, Instituto Mexicano del Seguro Social, Av. Lapislázuli 250, Col. El Haya, Villa de Álvarez 28984, Mexico; 10Escuela de Medicina, Plantel Guadalajara, Universidad Cuauhtémoc, Av. del Bajío No. 5901, Col. Del Bajío, Zapopan 45019, Mexico

**Keywords:** COVID-19, SARS-CoV-2, reinfection

## Abstract

Background: Repeated SARS-CoV-2 infections are plausible and related published data are scarce. We aimed to identify factors associated with the risk of recurrent (three episodes) laboratory-confirmed symptomatic SARS-CoV-2 infections. Methods: A retrospective cohort study was conducted, and 1,700 healthcare workers were enrolled. We used risk ratios (RR) and 95% confidence intervals (CI) to evaluate the factors associated with symptomatic SARS-CoV-2 infections. Results: We identified 14 participants with recurrent illness episodes. Therefore, the incidence rate was 8.5 per 10,000 person months. In a multiple-model study, vaccinated adults (vs. unvaccinated, RR = 1.05 [1.03–1.06]) and those with a severe first illness episode (vs. mild disease, RR = 1.05 [1.01–1.10]) were at increased risk for repeated symptomatic SARS-CoV-2 reinfections. Increasing age showed a protective effect (per each additional year of age: RR = 0.98 [0.97–0.99]). Conclusions: Our results suggest that recurrent SARS-CoV-2 infections are rare events in adults, and they seem to be determined, partially, by vaccination status and age.

## 1. Introduction

In Mexico, the burden of coronavirus disease 2019 (COVID-19) by severe acute respiratory syndrome coronavirus-2 (SARS-CoV-2) has been high [1]. Published data suggest that COVID-19 became the leading cause of death from 2020 to 2021 in Mexican territory [2].

The evidence derived from observational studies shows that patients who have recovered from coronavirus disease 2019 (COVID-19) by severe acute respiratory syndrome coronavirus-2 (SARS-CoV-2) can develop natural immunity [3,4]. However, high variability in antibody titers has been documented and healthcare workers seem to be at a high reinfection risk [5,6]. The reinfection risk in these specific population segments seemed to be higher during the dominance of the Omicron variant [7].

In addition, and even when preventing severe disease-related outcomes [8], waning immunity after COVID-19 vaccination has been documented in the months from when vaccination use became widespread [9,10,11,12]. A similar scenario has been documented after a natural SARS-CoV-2 infection [13].

Under these considerations, having repeated SARS-CoV-2 infections is plausible but related published data are scarce [14,15,16]. Assessing the frequency that reinfections occur, and the interval in which they occur after a previous illness episode is relevant from a clinical and epidemiological perspective to understand this phenomenon [17]. This study aimed to identify the factors associated with recurrent (three disease episodes) laboratory-confirmed symptomatic SARS-CoV-2 infections.

## 2. Materials and Methods

A retrospective and cohort study was conducted in a province of Mexico. The state of Colima (≈732 thousand inhabitants), where the study occurred, is in the western region of the country and on the central Pacific coast. Adults (aged 18 years or older) employed (as nurses, physicians, or others) by the Mexican Institute of Social Security (IMSS, the Spanish acronym) at any of its healthcare settings (n = 13) located across the state, were eligible. The IMSS is part of the Mexican health system and provides medical and care services to around 60% of the country’s total population.

Healthcare workers who recovered from a single laboratory-confirmed COVID-19 episode from March 2020 to February 2021 were eligible. The participants were integrated into the study sample at the date of recovery from the first illness episode and were followed until the start of a new COVID-19 episode, if applicable. The enrolled subjects contributed again to rate denominators once they fully recovered from the last disease episode or until 30 September 2022, when the follow-up ended.

The participants were classified as nurses (n = 378), physicians (n = 258), or other (n = 1064). This latter group was integrated, in general, by administrative or health personnel without direct patient care.

The main outcome was repeated symptomatic infection (three laboratory-confirmed COVID-19 episodes) by SARS-CoV-2 occurring during the study period (March 2020–September 2022; 30 months). Reinfection was defined using a positive RT-PCR or rapid antigen test in patients with respiratory symptoms, at least 90 days after full clinical recovery (disappearance of respiratory symptoms) from a COVID-19 episode [18]. The subjects with <90 days between the two positive tests were considered as persistent infections. 

The COVID-19 cases were identified by using data from an institutional system for the epidemiological surveillance of respiratory viral pathogens, which the use of is mandatory for the process of incapacity for work during illness recovery [19]. 

Clinical and epidemiological data of interest were retrieved from the medical files of the participants. The COVID-19 vaccination status, at enrollment, was evaluated. The vaccinated subjects were those with two shots of any COVID-19 vaccine or a single shot at ≥14 days before illness onset [20]. The BNT162b2 (Pfizer Inc./BioNTech, New York, NY, USA/Mainz, DE, Germany) started to be administered to healthcare workers in Mexico in the last week of January 2021. In the study sample, all the participants that were classified as immunized had received this vaccine.

The COVID-19 episodes were classified as non-severe and severe. These latter were those with clinical (fever or chills, cough, shortness of breath, and tachypnea) and radiographic findings (ground-glass patterns in X-ray or computed tomography imaging) of pneumonia that required hospital admission.

The participants were categorized according to the number of SARS-CoV-2 infections (1–2 vs. 3 episodes) and summary statistics were computed. We used risk ratios (RR) and 95% confidence intervals (CI), estimated using generalized linear regression models, to evaluate the factors associated with the risk of repeated symptomatic reinfection. This study was evaluated and approved by the Local Health Research Committee (601) of the Mexican Institute of Social Security (approval R-2022-601-024).

## 3. Results

A total of 1700 individuals were enrolled for a total follow-up of 16,474 person months. We identified 14 participants with three laboratory-positive COVID-19 episodes. Therefore, the overall rate of multiple symptomatic infections in the study sample was 8.5 per 10,000 person months. All the recurrent cases had no severe symptoms and hospital entry was needed.

The 32.1% of participants (n = 545/1700) were recruited based on a positive RT-PCR and the remaining (67.1%) were based on rapid antigen tests. The mean time (± standard deviation) to the second infection (first reinfection) was 292 ± 103 days, and the interval to the third infection was 131 ± 34 days.

When compared with subjects with two or fewer COVID-19 episodes, those with three episodes were younger (mean ± standard deviation: 30.8 ± 5.6 vs. 37.1 ± 8.1 years, *p* = 0.004) and were more likely to be fully COVID-19 vaccinated (57.1% vs. 8.5%, *p* < 0.001) at enrollment or to have had a severe first-episode (7.1% vs. 1.0%, *p* = 0.026). Most of the participants with recurrent infections (64.3%, n = 9/14) were administrative or health personnel without direct patient care. The remaining cases were nurses and none of them were physicians. A broader description of the study sample, according to the reinfection status, is presented in Table 1.

In the multiple regression model (Table 2), fully COVID-19-vaccinated adults (vs. unvaccinated, RR = 1.05 [1.03–1.06]) and those with a severe first illness episode (vs. mild disease, RR = 1.05 [1.01–1.10]) were at increased risk for repeated symptomatic SARS-CoV-2 reinfections. Increasing age showed a protective effect (per each additional year of age: RR = 0.98 [0.97–0.99]).

## 4. Discussion

We characterized a subset of Mexican healthcare workers with multiple symptomatic and laboratory-positive SARS-CoV-2 infections. Our results suggest that having three COVID-19 episodes was infrequent (about 8 cases per 1000) in the study sample and that vaccinated and younger participants were more likely to be reinfected.

A previously published study, and where healthcare professionals were enrolled, identified 11 cases of multiple SARS-CoV-2 reinfections (thrice infected) among 73 thousand participants [14]. In this study, the multiple reinfection cases were also young workers (mean age: 27 years; total range: from 22 to 56 years). The published data have documented age-mediated differences in COVID-19 risk-taking, and a lower risk perception for self and others has been observed in younger adults [21].

In another study that was conducted in Italy among healthcare workers, the risk for multiple reinfections (three episodes) was higher than ours (0.16% vs. 0.008%) [22]. We only tested for symptomatic cases and, in the Italian cohort, nearly a quarter (27.3%) of cases were asymptomatic. The latter may determine, at least partially, our heterogeneous estimates.

We also observed a 5% (95% CI 3–6%) increase in reinfection risk among vaccinated participants. The latter may be closely related to the high perceived efficacy of the COVID-19 vaccines that likely further reduced the adherence to other non-pharmaceutical interventions focusing on the prevention of respiratory viral spread. Thus, this may be an example of the Peltzman effect, where the risk perception decreases after safety measures have been implemented [23,24,25].

Interestingly, the individuals who recovered from the severe disease at primary infection showed a higher reinfection risk when compared with those who present mild symptoms (RR = 1.05, 95% CI 1.01–1.10). Similar findings have been published and may be related to increased perceived benefits from taking risks [26]. However, only 18 out of 1700 (1.1%) of the enrolled subjects had a severe first infection, so this finding must be considered with caution.

In our study, 9 out of 14 thrice-infected individuals had symptom onset during or after the dominance of the Omicron variant (B.1.1.529). A previously published study conducted among Mexican hospital workers also found a high reinfection rate during the dominance of this variant, even in fully vaccinated participants [27].

The proportion of vaccinated subjects in the study was low (9.1%) due to the time frame of enrollment (March 2020–February 2021) and the vaccination start in Mexican healthcare workers (January 2021). However, vaccine hesitancy was high in our country [28] and, together with lax interventions to limit social interaction, may have worsened the pandemic burden.

The potential limitations of the study must be cited. First, only workers from a specific healthcare institution were recruited. Therefore, even when their profiles are highly heterogeneous, they may not be entirely representative of the source population. Second, there is no genomic information for the infections. Thus, we were unable to determine which variant or subvariant was causing the infection, and a more infectious variant able to evade vaccine-derived or natural immunity [29] would lead to a higher prevalence of infection at the time of the epidemic wave caused by that variant.

Third, previously infected or immunized individuals may be less likely to develop symptomatic infections later (they may be reinfected but have a milder or asymptomatic illness) [30]. Consequently, they may not be tested, so these reinfections are not detected. Fourth, younger adults are more likely to have far more social contacts (and, therefore, potential sources of infection or opportunities to spread infection) than older adults [31]. This latter may also be determining the higher reinfection rates observed in younger participants. Fifth, we did not evaluate the long-term complications of COVID-19 in the study sample. Lung disease seems to be the most frequent complication among these patients [32] and it would have been clinically and epidemiologically useful to assess its risk among subjects with recurrent SARS-CoV-2 infections.

Sixth, about two-thirds of the participants were recruited based on a rapid antigen test result. Given that their sensitivity ranges from 72 to 80% [33], a subset of subjects integrating the study sample may be misclassified. Additionally, seventh, the vaccination status was only evaluated at enrollment. However, we must highlight that vaccine effectiveness decreased over time and according to the appearance of new SARS-CoV-2 variants and subvariants [34]. 

## 5. Conclusions

Our results suggest that recurrent SARS-CoV-2 infections are rare events and seem to be determined, at least partially, by vaccination status and age.

## Figures and Tables

**Table 1 vaccines-11-00626-t001:** Characteristics of the study sample for selected variables, Mexico 2020–2022.

Characteristic	Overall (n = 1700)		Number of COVID-19 Episodes				*p*
			One or Two (n = 1686)		Three (n = 14)		
Gender							
Female	1079	(63.5)	1069	(63.4)	10	(71.4)	0.535
Male	621	(36.5)	617	(36.6)	4	(28.6)	
Age (years) ^a^	37.1 ± 8.1		37.1 ± 8.1		30.8 ± 5.6		0.004
Age group							
<20	2	(0.1)	2	(0.1)	0	(0)	0.007
20–29	300	(17.7)	292	(17.3)	8	(57.2)	
30–39	826	(48.6)	821	(48.7)	5	(35.7)	
40–49	440	(25.8)	439	(26.0)	1	(7.1)	
50–59	119	(7.0)	119	(7.1)	0	(0)	
60+	13	(0.8)	13	(0.8)	0	(0)	
COVID-19 vaccination status ^b^							
Unvaccinated	1545	(90.9)	1539	(91.3)	6	(42.9)	<0.001
Vaccinated	155	(9.1)	147	(8.7)	8	(57.1)	
Severity of the first COVID-19 episode ^c^							
Mild	1682	(98.9)	1669	(99.0)	13	(92.9)	0.026
Severe	18	(1.1)	17	(1.0)	1	(7.1)	
Activity in healthcare services							
Other	1064	(62.6)	1055	(62.6)	9	(64.3)	0.192
Nurse	378	(22.2)	373	(22.1)	5	(35.7)	
Physician	258	(15.2)	258	(15.3)	0	(0)	
Personal history of:							
Obesity (BMI 30 or above)							
No	1451	(85.4)	1441	(85.5)	10	(71.4)	0.139
Yes	249	(14.6)	245	(14.5)	4	(28.6)	
Arterial hypertension							
No	1572	(92.5)	1559	(92.5)	13	(92.9)	0.956
Yes	128	(7.5)	127	(7.5)	1	(7.1)	
Type 2 diabetes mellitus							
No	1632	(96.0)	1618	(96.0)	14	(100)	0.443
Yes	68	(4.0)	68	(4.0)	0	(0)	
Asthma							
No	1647	(96.9)	1633	(96.9)	14	(100)	0.500
Yes	53	(3.1)	53	(3.1)	0	(0)	

Abbreviations: COVID-19, coronavirus disease 2019; BMI, body mass index. Note: The absolute and relative (%) frequencies are presented, except if others are specified. ^a^ Arithmetic mean ± standard deviation. ^b^ At enrollment. The vaccinated subjects were those with two vaccine shots or a single shot at ≥14 days before illness (second episode) onset. ^c^ Severe COVID-19 cases were those with clinical (fever or chills, cough, shortness of breath, and tachypnea) and radiographic findings (ground-glass patterns in X-ray or computed tomography imaging) of pneumonia that required hospital admission.

**Table 2 vaccines-11-00626-t002:** Factors associated with the risk of multiple (three or above) SARS-CoV-2 symptomatic reinfections, Mexico 2020–2022.

Characteristic	RR (95% CI), *p*			
	Bivariate Analysis		Multiple Analysis	
Gender				
Female	Ref.		Ref.	
Male	0.99 (0.98–1.01)	0.535	0.99 (0.98–1.01)	0.878
Age (years)	0.98 (0.97–0.99)	0.004	0.98 (0.97–0.99)	0.008
COVID-19 vaccination status ^b^				
Unvaccinated	Ref.		Ref.	
Vaccinated	1.05 (1.02–1.06)	<0.001	1.05 (1.03–1.06)	<0.001
Obesity (BMI 30 or above)				
No	Ref.		Ref.	
Yes	1.01 (0.99–1.03)	0.139	1.01 (0.99–1.02)	0.187
Severity of the first COVID-19 episode ^c^				
Mild	Ref.		Ref.	
Severe	1.05 (1.01–1.09)	0.025	1.05 (1.01–1.10)	0.020
Activity in healthcare services				
Other	Ref.		Ref.	
Nurse	1.00 (0.99–1.02)	0.378	1.01 (0.99–1.02)	0.434
Physician	0.99 (0.98–1.00)	0.177	0.99 (0.98–1.01)	0.467

Abbreviations: COVID-19, coronavirus disease 2019; BMI, body mass index. Note: The absolute and relative (%) frequencies are presented, except if other is specified. ^b^ At enrollment. The vaccinated subjects were those with two vaccine shots or a single shot at ≥14 days before illness (second episode) onset. ^c^ Severe COVID-19 cases were those with clinical (fever or chills, cough, shortness of breath, and tachypnea) and radiographic findings (ground-glass patterns in X-ray or computed tomography imaging) of pneumonia that required hospital admission.

## Data Availability

The data that support the findings of this study are not openly available and are available from the corresponding author upon reasonable request.

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
