# Peer review of "Predictors of Recurrent Laboratory-Confirmed Symptomatic SARS-CoV-2 Infections in a Cohort of Healthcare Workers"

_vaccines, 2023, doi:10.3390/vaccines11030626_

Round 1
Reviewer 1 Report
Vaccination and younger age are associated with recurrent laboratory-confirmed symptomatic SARS-CoV-2 infections
Comment: The title refers to the study's findings rather that stand alone tittle.
Why using younger age terminology as its read as <18 years you may say young adult for example. Advice to review the title and to include HCW and type of study
Methodology:
Participants were classified as nurses (n = 378), physicians (n = 258) or other (n = 79
1,064).
Comment: how the sample sizes were determined. Needed to be added in the methodology
A positive RT-PCR or rapid antigen test.
Comment: could you provide the results of how many cases were based RT-PCR and rapid antigen test??
Results:
The overall rate of multiple symptomatic infections in the study sample was 8.5 per 10,000 person-months.
Do you mean 8.5 per 10,000 person per months? Plz confirm!
Discussions
Lines 143-146In our study, 9 out of 14 thrice infected individuals had symptoms onset during or the after the dominance of the Omicron variant (B.1.1.529). A previously published study conducted among Mexican hospital workers also found a high reinfection rate during the 145 dominance of this variant, even in fully vaccinated participants [26].
Comment: Could also add other studies from other countries as well
Study limitation
Using the rapid antigen test could underestimation, could be one of the study limitation
Author Response
Comment 1. Vaccination and younger age are associated with recurrent laboratory-confirmed symptomatic SARS-CoV-2 infections. The title refers to the study's findings rather that stand alone tittle.
Response (R): We appreciate the thoughtful comments from Reviewer 1. We agreed and the new title of the manuscript is “Predictors of recurrent laboratory-confirmed symptomatic SARS-CoV-2 infections in a cohort of healthcare workers”.
Comment 2. Why using younger age terminology as its read as <18 years you may say young adult for example. Advice to review the title and to include HCW and type of study.
R: We agreed with Reviewer 1 about the terminology and “younger age” was omitted in the new version of the manuscript submitted for revision. The terms HCW and study design were included in the title of the new version of the manuscript submitted for revision.
Comment 3: Methodology: Participants were classified as nurses (n = 378), physicians (n = 258) or other (n = 79). How the sample sizes were determined. Needed to be added in the methodology.
R: Healthcare workers who recovered from a first laboratory-confirmed COVID-19 episode from March 2020 to February 2021, were eligible. This is now included in the new version of the manuscript submitted for revision (please see lines 77-78).
Comment 4: A positive RT-PCR or rapid antigen test. Comment: could you provide the results of how many cases were based RT-PCR and rapid antigen test??
R: Yes. A total of 545 participants (32.1%) were based on RT-PCR and the remaining (67.1%, n = 1,155) were based on rapid antigen tests. This is now included in the new version of the manuscript submitted for revision (please see lines 118-119).
Comment 5: Results: The overall rate of multiple symptomatic infections in the study sample was 8.5 per 10,000 person-months. Do you mean 8.5 per 10,000 person per months? Plz confirm!
R: We believe that how it is currently presented (8.5 per 10,000 person-months) is adequate since we refer to person-time.
Comment 6: Discussions. Lines 143-146In our study, 9 out of 14 thrice infected individuals had symptoms onset during or the after the dominance of the Omicron variant (B.1.1.529). A previously published study conducted among Mexican hospital workers also found a high reinfection rate during the 145 dominance of this variant, even in fully vaccinated participants [26]. Comment: Could also add other studies from other countries as well
R: Unfortunately, similar published data are scarce. However, we were able to find another recent study that was conducted in Italy and its findings are briefly discussed in the new version of the manuscript submitted for revision (please see lines 150-154).
Comment 7: Study limitation. Using the rapid antigen test could underestimation, could be one of the study limitation.
R: We agreed with Reviewer 1 and this latter is now included as a study limitation in the new version of the manuscript submitted for revision (please see lines 193-195).
Reviewer 2 Report
The manuscript of Ríos-Silva et al reports on a retrospective analysis in 1,700 health care workers of the State of Colima Mexico employed in 13 healthcare locations during the period of March 2020 up to September 30, 2022, for recurrent SARS.CoV-2 infections.
The study aims to identify possible risks factors associated with risk of recurrent infections, defined as three episodes of confirmed (PCR or antigen test) infection in patients with symptomatic respiratory symptoms, with a distance of at least 90 days from last positive testing.
Comments:
The number of vaccinated people over the total analysed population is of 155/1,700. In the first period of this retrospective study no vaccine for Covid-19 was yet approved. This needs to be mentioned and the impact on the study results discussed. Since not all the vaccines offers the same protections, the type of vaccine used needs to be mentioned (suggest including this information in Table 1). How was a person considered fully vaccinated? Had all the people that were listed as vaccinated completed the full vaccination protocol including booster shots according to the respective vaccines used?
In several Countries around the world restrictions were put in place for non-vaccinated people. Where there any restrictions for unvaccinated people in Mexico, and what could have been the impact on risk of reinfection (limited social interaction)?
Definition of mild and severe Covid-19 episode needs to be included.
It is reported that patients with a first severe episode had a higher risk of having 3 infections. 1 in 18 patients that had a severe first infection had 3 infections, versus 13 in 1682 that had a first mild infection. Even if statistically correct, are the numbers sufficient to draw this conclusion?
Where there peoples inserted in the study who had the second infection at about 90 days before the end of study follow-up?
The ratio of vaccinated versus unvaccinated in the studies population seems overall different to what reported in other regions, e.g. Europe or U.S.A. were a high rate of vaccination is reported, especially among health workers. Please comment on this in the discussion
Please complete the title with the characteristic of the patent population studied and the location were the study was performed. Such as Vaccination and younger age are associated with recurrent laboratory confirmed symptomatic SARS-Cov-2 infections among Health care workers in the State of Colima, Mexico.
Author Response
The manuscript of Ríos-Silva et al reports on a retrospective analysis in 1,700 health care workers of the State of Colima Mexico employed in 13 healthcare locations during the period of March 2020 up to September 30, 2022, for recurrent SARS.CoV-2 infections.
The study aims to identify possible risks factors associated with risk of recurrent infections, defined as three episodes of confirmed (PCR or antigen test) infection in patients with symptomatic respiratory symptoms, with a distance of at least 90 days from last positive testing.
Comments:
Comment 1: The number of vaccinated people over the total analyzed population is of 155/1,700. In the first period of this retrospective study no vaccine for Covid-19 was yet approved. This needs to be mentioned and the impact on the study results discussed. Since not all the vaccines offers the same protections, the type of vaccine used needs to be mentioned (suggest including this information in Table 1). How was a person considered fully vaccinated? Had all the people that were listed as vaccinated completed the full vaccination protocol including booster shots according to the respective vaccines used?
Response (R): We appreciate the thoughtful comments from Reviewer 2. The COVID-19 vaccination status, at enrollment, was evaluated. The vaccinated subjects were those with two shots of any COVID-19 vaccine or a single shot at ≥ 14 days before illness onset. The BNT162b2 (Pfizer, Inc./BioNTech) started to be administered to healthcare workers in Mexico in the last week of January 2021. In the study sample, all the participants that were classified as immunized had received this vaccine. This is now included in the new version of the manuscript submitted for revision (please see lines 97-101). It is also included in Table 1 as suggested.
Comment 2: In several Countries around the world restrictions were put in place for non-vaccinated people. Where there any restrictions for unvaccinated people in Mexico, and what could have been the impact on risk of reinfection (limited social interaction)?
R: No restrictions were applied in Mexico to unvaccinated subjects. This latter, together with vaccine hesitancy, may have worsened the disease burden which was high in the Mexican population. This is now included in the new version of the manuscript submitted for revision (please see lines 171-175).
Comment 3: Definition of mild and severe Covid-19 episode needs to be included.
R: Severe COVID-19 cases were those with clinical (fever or chills, cough, shortness of breath, and tachypnea) and radiographic findings (ground-glass patterns in x-ray or computed tomography imaging) of pneumonia that required hospital admission. This is now included in the new version of the manuscript submitted for revision (please see lines 102-105) and in Tables 1 and 2.
Comment 4: It is reported that patients with a first severe episode had a higher risk of having 3 infections. 1 in 18 patients that had a severe first infection had 3 infections, versus 13 in 1682 that had a first mild infection. Even if statistically correct, are the numbers sufficient to draw this conclusion?
R: We agree with Reviewer 2. In the new version of the manuscript submitted for revision, we briefly discuss the relevance of considering this finding with caution (please see lines 164-166).
Comment 5: Where there peoples inserted in the study who had the second infection at about 90 days before the end of study follow-up?
R: They were considered as second infection due to the administrative reasons of the study protocol.
Comment 6: The ratio of vaccinated versus unvaccinated in the studies population seems overall different to what reported in other regions, e.g. Europe or U.S.A. were a high rate of vaccination is reported, especially among health workers. Please comment on this in the discussion
R: The proportion of vaccinated subjects in the study was low (9.1%) due to the time framework of enrollment (March 2020 – February 2021) and the vaccination start in Mexican healthcare workers (January 2021). This is now included in the new version of the manuscript submitted for revision (please see lines 171-175).
Comment 7: Please complete the title with the characteristic of the patent population studied and the location were the study was performed. Such as Vaccination and younger age are associated with recurrent laboratory confirmed symptomatic SARS-Cov-2 infections among Health care workers in the State of Colima, Mexico
R: We agree with Reviewer 2 and the title of the manuscript was modified as suggested.
Round 2
Reviewer 2 Report
The authors have adressed all the concerns and request of clarifications, raised in the initial review.